# NAHAS: Neural Architecture and Hardware Accelerator Search

## Abstract

Neural architectures and hardware accelerators have been two driving forces for the rapid progress in deep learning. Although previous works have optimized either neural architectures given fixed hardware, or hardware given fixed neural architectures, none has considered optimizing them jointly. In this paper, we study the importance of co-designing neural architectures and hardware accelerators. To this end, we propose NAHAS, an automated hardware design paradigm that jointly searches for the best configuration for both neural architecture and accelerator. In NAHAS, accelerator hardware design is conditioned on the dynamically explored neural networks for the targeted application, instead of fixed architectures, thus providing better performance opportunities. Our experiments with an industry-standard edge accelerator show that NAHAS consistently outperforms previous platform-aware neural architecture search and state-of-the-art EfficientNet on all latency targets by 0.5% - 1% ImageNet top-1 accuracy, while reducing latency by about 20%. Joint optimization reduces the search samples by 2x and reduces the latency constraint violations from 3 violations to 1 violation per 4 searches, compared to independently optimizing the two sub spaces.

## 1 Introduction

Conventional hardware design has been driven by benchmarks (e.g. SPEC (SPE)) where a selected set of workloads are evaluated and the average performance is optimized. For example, CPUs are optimized for sequential workloads such as desktop applications and GPUs are designed for massive parallel workloads such as gaming, graphics rendering, scientific computing, etc. Generalization over a wide set of representative workloads is the traditional method for hardware design and optimization. However, the selected workloads can stay fixed for a substantially long time, which makes the hardware design lag behind the algorithmic changes.

As a result of hitting the end of Moore's Law in the recent decade, the focus has switched to hardware specialization to provide additional speedups and efficiency for a narrowed-down application or domain. Google's TPU (TPU) and Intel's Nirvana NNP (Yang, 2019) are two representative accelerators specialized for deep learning primitives and MLPerf (MLP) has become prevalent for benchmarking the state-of-the-art design of ML accelerators. However, rapid progress in deep learning has given birth to numerous more powerful, expressive, and efficient models in a short time, which results in both benchmarking and accelerator development lagging behind. For example, squeeze-and-excite with global pooling and SiLU/Swish non-linearity (Ramachandran et al., 2017; Elfwing et al., 2018) are found to be useful in EfficientNet (Tan & Le, 2019), however, neither can currently execute efficiently even on a highly specialized accelerator. We need to evolve the accelerator design more rapidly.

On the other hand, platform-aware neural architecture search (Tan et al., 2019; Wu et al., 2019; Cai et al., 2018) optimizes the neural architectures for a target inference device. The target device has a fixed hardware configuration that can significantly limit NAS flexibility and performance. For example, the target device may have a sub-optimal compute and memory ratio for the target application combined with a inference latency target, which can shift the optimal NAS model distributions and result in underperformance.

We propose NAHAS, a new paradigm of software and hardware co-design, by parameterizing neural architecture search and hardware accelerator search in a unified joint search space. We use a highly parameterized industry-standard ML accelerator as our target device, which has a tunable set of important hardware parameters. These knobs fundamentally determine hardware characteristics such as number of compute units, amount of parallelism, compute to memory ratio, bandwidth, etc., which we found very critical to model performance. We formulate the optimization problem as a bi-level optimization with hardware resource constraints on chip area and model latency.

Unlike conventional hardware optimization, NAHAS is a *task driven* approach, where the task is a problem (e.g. image classification, object detection) or a domain of problems (e.g. vision, NLP), not a set of fixed programs or graphs (e.g. ResNet, Transformers). This effectively creates generalization across the vertical stack, making the hardware evolve with the applications. NAHAS can be practically used to design customized accelerators for autonomous driving and mobile SoC (system-on-chip) where a set of highly optimized accelerators are combined into a system.

We also propose a latency-driven optimization that maximizes model accuracy while meeting a latency constraint under a chip area budget. Conventional platform-aware NAS typically focuses on searching efficient NAS models with higher accuracy, lower parameters and FLOPs (number of multiply-add operations). However, optimizing for lower parameters and FLOPs is not necessarily good for performance (Wu et al., 2019). For example, a multi-branch network like NasNet (Zoph et al., 2017) has lower parameters and FLOPs compared to layer-wise network such as ResNet and MobileNet, but its fragmented cell-level structure is not hardware friendly and can be evaluated very slowly on the target device. As a matter of fact, the number of parameters and FLOPs are only *indirect metrics* that could affect a direct metric such as latency and power negatively if not considered. For example, the model can run out of memory if the number of parameters is too large. However, a direct optimization on *indirect metrics* won't necessarily improve the *direct metrics*.

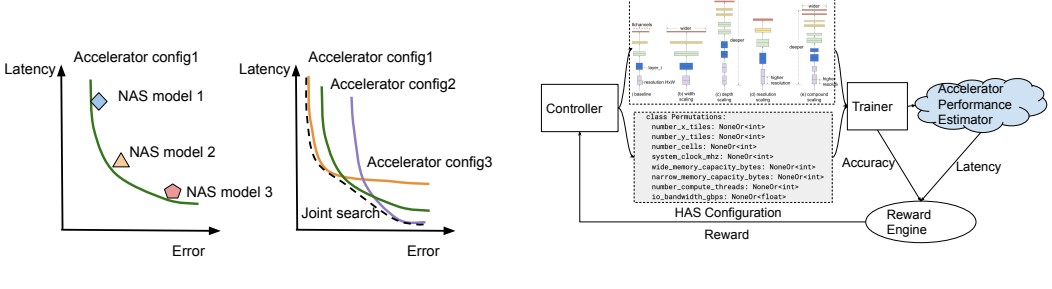

**(a)** NAHAS Extends the Pareto Frontier  **(b)** NAHAS Work Flow

**Figure 1:** Motivating example (a) and high level work flow (b). In (a) different accelerator configurations have different pareto frontiers consisting of different NAS models (left) and joint search effectively extends the pareto frontier by joining multiple frontiers (right).

Figure 1 shows a motivating example of using joint search and a high level workflow of NAHAS. While conventional platform-aware NAS selects models along the pareto frontier with different latency and accuracy tradeoffs for one target device, as indicated in Figure 1a left, NAHAS further expands the pareto frontier by enabling different hardware accelerator configurations. To summarize our contributions:

- We develop a fully automated framework that can jointly optimize neural architectures and hardware accelerators. For the first time, we demonstrate the effectiveness of co-optimizing neural architecture search with the parameterization of a highly optimized industry-standard accelerator.

- We propose a latency-driven search method, which is hardware-agnostic and achieves state-of-the-art results across multiple search spaces. NAHAS outperforms MnasNet and EfficientNet on all latency targets by 0.5%-1% in ImageNet accuracy and 20% in latency.

- We observe that different model sizes combined with different latency targets require completely different hardware accelerator configurations. Customizing accelerators for different model sizes and latency targets becomes essential when co-designing neural architectures and accelerators for domains such as autonomous driving.

- We compare different optimization strategies and find that joint search consistently outperforms optimizing NAS and HAS in an alternating fashion. Oneshot search can significantly reduce search cost (total number of samples and search time), however, it is less suitable for large models when constructing the super network is too costly.

## 2 RELATED WORKS

**ML-driven architecture search:** Design space exploration in computer systems has become more crucial due to the surge of specialized hardware (Parsa et al., 2020; Iqbal et al., 2020; Sun et al., 2020; Nardi et al., 2019; Cong et al., 2018; Koeplinger et al., 2018; Balaprakash et al., 2016; Ansel et al., 2014). Hierarchical-PABO (Parsa et al., 2020) and FlexiBO (Iqbal et al., 2020) use multi-objective Bayesian optimization for neural network accelerator design. In order to reduce computational cost, (Sun et al., 2020) apply genetic algorithm to design CNN models without modifying the underlying architecture. HyperMapper (Nardi et al., 2019) uses random forest in automatic tuning of hardware accelerator parameters in a multi-objective setting.

**Platform-aware neural architecture search:** MnasNet (Tan et al., 2019) pioneered platform-aware neural architecture search where for the first time NAS models are tailored for a target hardware device and resource-efficient models are identified. ProxylessNAS (Cai et al., 2018) directly learns the architectures for ImageNet by proposing a gradient-based approach to train binarized parameters. FBNet (Wu et al., 2019) proposes a differentiable platform-aware NAS using Gumbel Softmax and ChamNet (Dai et al., 2019) proposes a neural architecture adaptation method using efficient accuracy and resource predictors. Accelerator-aware NAS (Gupta & Akin, 2020) for the first time targets industry-standard accelerators and identified SoTA models for an edge TPU. However, none of these work optimizes the underlying hardware accelerators together with NAS.

**Co-design:** There is a growing body of work exploring neural architecture search and hardware design (Jiang et al., 2020a; Choi et al., 2020; Jiang et al., 2020b; Achararit et al., 2020; Yang et al., 2020; Kwon et al., 2018; Yang et al., 2018). However, most of the work target FPGAs or academic accelerators such as Eyeriss Chen et al. (2016), where the hardware is less optimized on real workloads (e.g. ImageNet) and the performance model is less accurate. NAHAS, however, targets a highly optimized industry-standard ML accelerator and the performance simulator is cycle-accurate that has been validated. Moreover, NAHAS demonstrates effectiveness on real ImageNet workload, outperforming SoTA models on two different search spaces.

## 3 METHOD

In this section, we will formulate the joint optimization problem of NAHAS and present two different approaches.

### 3.1 FORMULATION

The objective for NAHAS it to find a neural architecture parameter $\alpha$ and hardware accelerator parameter $h$ such that the validation accuracy on a ImageNet classification task can be maximized while meeting a chip area and latency target.

$$\min_{\alpha,h} \mathcal{L}(\alpha, h, w_\alpha^*, \mathbb{D}_{val}) \; s.t. \; w_\alpha^* = \arg\min_{w_\alpha} \mathcal{L}(\alpha, h, w_\alpha, \mathbb{D}_{train})$$

$$Latency(\alpha, h) \leq T_{latency}, \; Area(h) \leq T_{area}.$$

where $\mathcal{L}$ indicates the objective function of the tasks (e.g., cross-entropy for classification) and $w_\alpha$ denotes the weights of the architecture $\alpha$. NAHAS introduces a broader search space than NAS (neural architecture search) or HAS (hardware accelerator search) alone, with the flexibility to fix either $\alpha$ or $h$ therefore, the optimization problem is reduced to NAS or HAS. We empirically compared different optimization strategies in Section 4.4.

### 3.2 NAS SEARCH SPACE

**MobileNetV2:** We build the architecture search space $\mathcal{S}_1$ based on the standard MobileNetV2. The search space is tailored for mobile edge processors, therefore, consists mostly of efficient operations

such as mobile inverted bottleneck convolution (MBConv). Specifically, we search for the kernel size from {3, 5, 7} for each MBConv, and we also search for the expansion ratio from {3, 6} for each block except for the first one, which has the default expansion ratio of 1. In MobileNetV2, there are 17 inverted residual blocks, and thus the cardinality of $\mathcal{S}_1$ is about 8.4e12.

**EfficientNet:** In order to create larger NAS models and to better leverage modern edge accelerators which has larger number of compute units and memory capacities, we build the architecture search space $\mathcal{S}_2$ based on the standard EfficientNet-B0. Similar to $\mathcal{S}_1$, we also search for the kernel size from {3, 5, 7} and the expansion ratio from {3, 6}. Since there are 16 inverted residual blocks in EfficientNet-B0, the cardinality of $\mathcal{S}_2$ is about 1.4e12.

### 3.3 HAS Search Space

The target device is an industry-standard, highly parameterized edge accelerator which allows us to create various configurations in a large design space with tradeoffs between performance, power, area, and cost. The accelerator features a set of parallel processing elements (PE) organized in a 2D tile. The number of PEs in each dimension determines the aspect ratio of the chip. In each PE there are multiple compute lanes that shares a local memory and each lane has a register file and a series of single-instruction multiple-data (SIMD) style multiply-accumulate (MAC) compute units. Each of these architecture components provide a degree of parallelism and a corresponding area cost. With a fixed chip area budget, HAS optimizes the on-chip resource allocation, balancing the compute and memory and searching for the best chip parameterization for a given application.

**Table 1:** Edge Accelerator Search Space

| parameters | type | search space | parameters | type | search space |
|---|---|---|---|---|---|
| PEs_in_x_dimension | int | 1, 2, 4, 6, 8 | local_memory_MB | int | 0.5, 1, 2, 3, 4 |
| PEs_in_y_dimension | int | 1, 2, 4, 6, 8 | compute_lanes | int | 1, 2, 4, 8 |
| SIMD_units | int | 16, 32, 64, 128 | io_bandwidth_gbps | float | 5, 10, 15, 20, 25 |
| register_file_KB | int | 8, 16, 32, 64, 128 | | | |

As baseline we take the default accelerator configuration which is optimized while considering a series of production workloads from multiple domains. The baseline configuration features 4x4 PEs where each PE has 2 MB local memory and 4 compute lanes. Each compute lane has a 32 KB register file and 64 4-way SIMD units. Since the accelerator is targeted for edge use cases, SIMD units can sustain the peak throughput for 8-bit quantized operations. This baseline configuration can deliver a peak throughput of 26 TOPS/s at 0.8 GHz.

Unlike the NAS search space, the HAS search space contains many invalid points, which makes training a cost model or joint search with the in-house simulator more challenging. Invalid configurations can be caused by many reasons. For example, the created accelerator configuration in combined with the NAS model is not supported by the compiler or a NAS model is created too large for the generated HAS configuration, etc.

### 3.4 Search Objective

While power and area can be other important metrics to be considered for efficient accelerator design, we focus on maximizing model accuracy while meeting an inference latency constraint on a target device. We also impose a chip area constraint, which is set equal to the baseline accelerator design. The paper does not minimize chip area or latency, as in practise they don't need to be minimized. However, the chip resource constraint can be set to area, power, energy, or a combination of many, however, this paper focuses on an area constraint only.

Similar to Mnasnet (Tan et al., 2019), we use a customized weighted product to encourage Pareto optimal solutions. More specifically, we have a *optimization metric* of model accuracy and two *hardware constrain metrics* of model latency and chip area. The optimization goal is to maximize model accuracy while meeting the latency and area constraints.

$$\max_{\alpha,h} Accuracy(\alpha,h) \times \left[\frac{Latency(\alpha,h)}{T_{latency}}\right]^{w_0} \times \left[\frac{Area(h)}{T_{area}}\right]^{w_1}$$

where $w_0, w_1$ are the weight factors:

$$w_0 = \begin{cases} p, & \text{if } Latency(\alpha, h) \leq T_{latency} \\ q, & \text{otherwise} \end{cases} \qquad w_1 = \begin{cases} p, & \text{if } Area(h) \leq T_{area} \\ q, & \text{otherwise} \end{cases}$$

When $p = 0, q = -1$, the reward function imposes a hard latency constraint and we simply use accuracy as the objective if the measured latency is meeting the latency target $T$ and only sharply penalize the objective value if the sample violates the latency constraint. When $p = q = -0.07$[1], the reward becomes a soft constraint function. In the NAHAS evaluations, we use both rewards for different experiments.

## 3.5    Optimizing the Joint Search Space Without Weight Sharing

Similarly as NASNet (Zoph et al., 2017) and MNASNet (Tan et al., 2019), we use PPO (Schulman et al., 2017) as the controller algorithm to optimize the joint search space from NAS and HAS. The controller samples the search space using a recurrent network, each sample is trained by a child program. For the MobileNetV2 search space, we use a proxy task that trains each sample on ImageNet for only 5 epochs and it takes 5000 samples for the controller to converge. For larger models using EfficientNet search space, we find that training the proxy task for 15 epochs while reducing the total number of samples to 2000 improves the results.

## 3.6    Optimizing the Joint Search Space With Weight Sharing

To further reduce the search cost, we employ an efficient search method with weight sharing. Similarly as ProxylessNAS (Cai et al., 2018) and TuNAS (Bender et al., 2020), we use the controller decisions from the NAS space to construct a super-network for optimizing the architecture, meanwhile using the decisions from the HAS space to create a sub-graph for computing the cost. Decision points from both spaces are optimized by a RL algorithm within the same graph. For each training step, we train the model weights and the controller decision points in an interleaved way. To achieve better results, we apply the absolute reward function and RL warm-up procedure introduced in TuNAS.

To estimate the latency of the model on a given accelerator configuration, we train a cost model with random generated samples using an in-house accelerator simulator. We need a cost model because as NAS becomes much faster with oneshot search, the query to the accelerator performance simulator for chip area and inference latency becomes the new bottleneck for NAHAS oneshot search. Given a sampled neural architecture configuration and an accelerator configuration, the cost model predicts the accelerator area $f_a(h)$ and model accuracy $f_l(\alpha, h)$. We use a MLP network with ReLU to encapsulate the non-linearity in the latency prediction. The area predictor and latency predictor largely share parameters with only separate parameterization in the prediction heads.

$$Loss = MSE(L_a, f_a(h)) + \lambda MSE(L_l, f_l(\alpha, h))$$

The cost model was trained with 500k labeled data randomly generated by permuting the neural architecture configurations and accelerator configurations. We use a 3-layer MLP of hidden size 256 and apply a dropout of 0.1 to mitigate overfitting at each layer.

## 4    Evaluation

### 4.1    Experiment Setup

**Accelerator Performance Simulator:** Evaluating the candidate neural and hardware architectures accurately is a key requirement for the NAHAS framework. Simply using number of MACs or parameters of the neural model as a proxy for performance can be highly inaccurate as their performance highly depends on how the neural network is mapped on the hardware architecture and their unique compute characteristics. To this end, we have utilized an in-house cycle-accurate performance simulator and an analytical area model based on hardware synthesis. We deployed

---

[1]Tan et al. (2019) found -0.07 to empirically ensure Pareto-optimal solutions have similar reward under different accuracy-latency trade-offs.

both of these estimators as a service where multiple NAHAS clients can send parallel requests. This provides a flexible way to scale-up the performance and area evaluation tasks.

**Search Hyperparameters:** For the end2end search, we choose PPO as it is tested by time. We use the average performance of 10 trials as a reward. We use Adam optimizer with the learning rate of 0.0005 to update the controller, where the policy gradients are clipped by 1.0. For each trial, we train the sampled candidate by five epochs using RMSProp. For these five epochs, we will first warm up the model by two epochs using the learning rate from 0 to 0.66, and then cosine decay it from 0.66 to 0 for the rest three epochs. For the oneshot search, we utilize REINFORCE to optimize the controller following TuNAS. We use Adam with a learning rate of 0.0048 to optimize it and use the momentum as 0.95 for baseline. In addition, we use RMSProp to optimize the shared weights following the same learning rate schedule as TuNAS. The latency predictor is pre-trained.

**Cost Model Hyperparameters:** The cost model described in Section 3.6 was trained with hyperparameters defined in Table 2. In the oneshot search, we replace the accelerator performance simulator with the trained cost model and search for the best model for five different latency targets (0.3ms, 0.5ms, 0.8ms, 1.1ms, and 1.3ms). The average error between the latency target and the estimated latency of the best model using the accelerator performance simulator is only 0.4%.

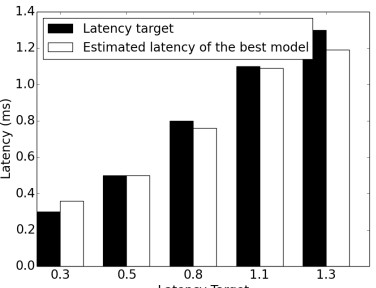

**Figure 2:** Cost Model Accuracy.

| Optimizer | Adam |
|---|---|
| Loss Re-weight $\lambda$ | 10 |
| Batch size | 128 |
| Hidden dimension | 256 |
| Learning rate | 0.001 |
| Training steps | 600k |
| Input feature size | 394 |

**Table 2:** Cost Model Hyperparameters.

## 4.2 SAMPLE DISTRIBUTIONS

To study the sample distribution during search, we first compare our NAHAS search with previous platform-aware NAS, as shown in Figure 3a and Figure 3b. Search is performed on two search spaces: EfficientNet-B0 (relatively small) and EfficientNet-B1 (relatively large). In platform-aware NAS, the target device is fixed to the baseline accelerator design, as described in Section 3.3; whereas in NAHAS search, the target chip area is set to the same as the target device of platform-aware search. Latency targets are set of 1ms and 2ms for these two search spaces. We observe that without the flexibility to change the hardware configuration, platform-aware NAS always converges to sub-optimal solutions of either higher latency or lower accuracy. For NAHAS, not all the samples traversed meet the chip area constraint. However, traversing through samples violating the resource constraints (red points in the figure) can help converge to more pareto-optimal samples eventually with both higher accuracy and lower inference latency.

Figure 3c and Figure 3d also compare two variants of NAHAS: joint search and phase search. In a joint NAHAS search, the two search spaces (NAS and HAS) are unified and a sample is taken from the joined search spaces. However in a phase NAHAS search, we fix one search space of the two while searching the other, aiming for a similar boost in optimization performance similar to using coordinate descent. Empirically, we observe that with the same number of samples, NAHAS joint search consistently outperforms NAHAS phase search, producing models with higher accuracy and lower latency. It is possible that fixing one search space results in a loss surface with more saddle points such that the optimization algorithms can easily get stuck.

In summary, Figure 3 demonstrates the intuition of the selection of a NAHAS joint search approach.

## 4.3 NAHAS SEARCH RESULTS

We compare NAHAS with MnasNet, Fixed hardware NAS (NAHAS with default accelerator configuration), and EfficientNetB0-B2, by scaling the latency targets from 0.3ms (on the smaller MobileNetV2 search space) to 2.0ms (on the larger EfficientNet search space), and find that NAHAS

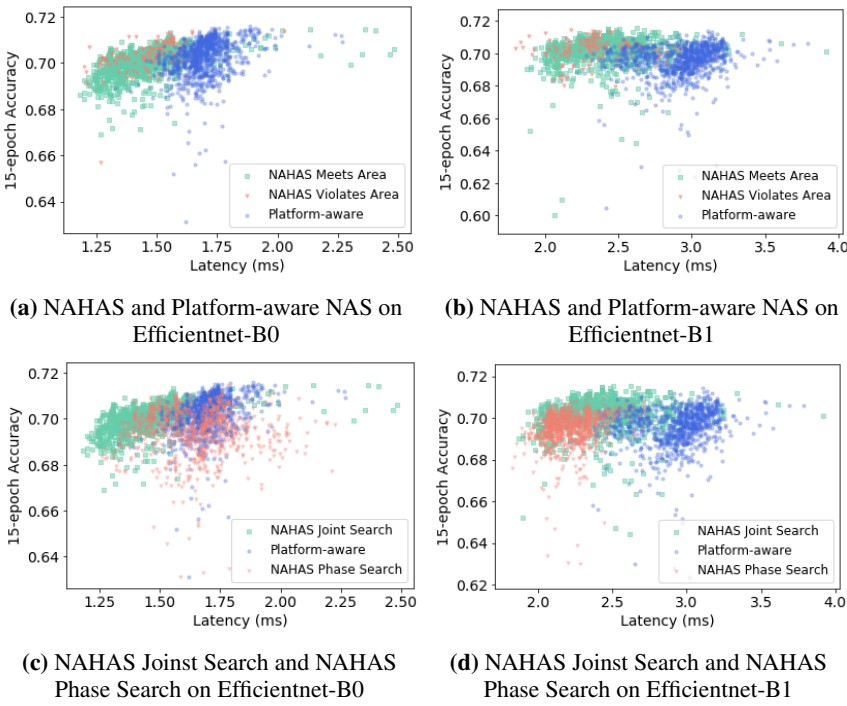

**(a)** NAHAS and Platform-aware NAS on Efficientnet-B0

**(b)** NAHAS and Platform-aware NAS on Efficientnet-B1

**(c)** NAHAS Joinst Search and NAHAS Phase Search on Efficientnet-B0

**(d)** NAHAS Joinst Search and NAHAS Phase Search on Efficientnet-B1

**Figure 3:** Searched sample distributions comparisons.

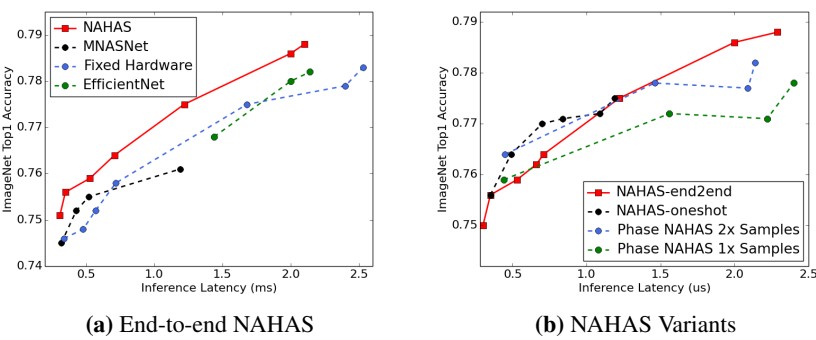

**(a)** End-to-end NAHAS

**(b)** NAHAS Variants

**Figure 4:** ImageNet Top1 Accuracy vs. Inference Latency.

is about 0.5% to 1% better in accuracy at every latency target, as shown in Figure 4a. The lessons we learnt from this experiments include:

- Changing the hardware accelerator configurations helps identify higher accuracy models that cannot be identified by a fixed hardware NAS.
- Larger models (e.g. EfficientNetB2) requires a higher memory-to-compute ratio in the accelerator design, compared to smaller models (e.g. MobileNetV2 or ProxylessNAS).
- Customizing the hardware accelerator for different model sizes and latency targets is beneficial to model performance and hardware efficiency.

In addition to the accuracy gains, we also observe latency reduction and chip area reduction when co-optimizing the neural architectures and hardware accelerators, as shown in Table 3.

- Oneshot search is more effective than end-to-end search for smaller models with lower than 1ms latency. NAHAS-oneshot can identify models 2% better in accuracy than MnasNet with a lower latency.
- NAHAS can identify accelerator configurations of much lower area, compared to the baseline default. Apart from improving model accuracy and latency, NAHAS can be applied to reduce chip area and power consumption (with a different reward).

- Phase-NAHAS performance depends on the initial fixed NAS configuration and can often lead to constraint violation. For example, Phase-NAHAS identifies models of 1% higher accuracy than EfficientNetB0, however, it violates the latency constraint.

**Table 3:** Comparison on accuracy v.s. latency with previous approaches. Models are grouped to different regimes and sorted by accuracy. On the fast latency regime, NAHAS-oneshot has the best performance, while on the high accuracy regime, NAHAS-end2end achieves the best performance.

| Model | Top-1 Acc. | Latency (us) | Normalized chip area |
|---|---|---|---|
| MobileNetV2 (Sandler et al., 2018) | 74.4% | 0.3 | 1.0 |
| Mnasnet-B1 (Tan et al., 2019) | 74.5% | 0.41 | 1.0 |
| ProxylessNAS (Cai et al., 2018) | 74.8% | 0.42 | 1.0 |
| NAHAS-end2end | **74.9%** | 0.3 | 1.0 |
| NAHAS-oneshot | **76.5%** | 0.35 | 0.52 |
| Mnasnet-D1 (Tan et al., 2019) | 75.1% | 0.51 | 1.0 |
| NAHAS-end2end-0 | **75.1%** | 0.47 | 0.5 |
| NAHAS-end2end-1 | **75.4%** | 0.48 | 1.0 |
| NAHAS-end2end-2 | **76.1%** | 0.66 | 1.0 |
| Phase-NAHAS-end2end | **76.4%** | 0.45 | 1.4 |
| NANAS-oneshot | **76.8%** | 0.49 | 0.64 |
| EfficientNet-B0 (Tan & Le, 2019) | 76.8% | 1.44 | 1.0 |
| NAHAS-oneshot-0 | **77.2%** | 1.09 | 0.78 |
| NAHAS-oneshot-1 | **77.4%** | 1.19 | 0.78 |
| NAHAS-end2end | **77.4%** | 1.2 | 1.0 |
| Phase-NAHAS-end2end | **77.8%** | 1.40 | 1.0 |
| EfficientNet-B1 (Tan & Le, 2019) | 78% | 2.0 | 1.0 |
| Phase-NAHAS-end2end | **77.7%** | 2.0 | 0.93 |
| NAHAS-end2end | **78.6%** | 2.0 | 0.93 |

## 4.4 OPTIMIZATION STRATEGIES

Figure 4b compares variants of NAHAS. More particularly, we compare the end-to-end NAHAS joint search with oneshot search, and phase-based NAHAS. In a phase-based NAHAS, we start with a HAS on a fixed initial neural architecture in the searc-h space (Mobil-eNetV2, EfficientNet-B0, EfficientNet-B1, and EfficientNet-B2) with a soft constraint function described in Section 3.4, aiming to find a accelerator configuration which is pareto-optimal in terms of latency and chip area. Then we apply a NAS with a hard constraint function described in Section 3.4 on the selected best accelerator configuration, aiming to identify good neural architectures that strictly meet the hardware latency constrain. We compare phase NAHAS with 1x and 2x total searched samples compared to the NAHAS end-to-end joint search baseline.

Comparing end-to-end with oneshot, we find that oneshot is more effective for smaller models with lower latencies where constructing a super network is more practical. Oneshot NAHAS reduces the search cost from 32 TPU-days to 1 TPU-day for a MobileNetV2 search space. However, oneshot is not suitable for large models such as models in EfficientNet search spaces.

NAHAS phase search with the same number of searched samples performs much worse than the NAHAS end-to-end joint search. Doubling the total searched samples (total search time) improves the quality of results. However, depending on the search space, latency target, and the initial accelerator configuration, performance varies significantly using NAHAS phase search.

## 5 CONCLUSION

We propose NAHAS, a software/hardware co-design that jointly optimizes neural architecture search and hardware accelerator search on a industry-standard ML accelerator. Jointly optimizing the application and hardware expands the Pareto frontier that enables more accurate models for any given latency targets. Moreover, it enables more rapid evolution of hardware along with the software stack.

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
