# OpenReview forum: "NAHAS: Neural Architecture and Hardware Accelerator Search"
_ICLR.cc/2021/Conference — Reject_

### Official Review · AnonReviewer2 · 2020-10-26
**The paper needs to improve its clarity**

**Rating:** 6
**Confidence:** 4

**Review:**

##########################################################################

Summary:

The paper presents NAHAS, which is a combination of Neural Architecture Search (NAS) and Hardware Architecture Search (HAS)
for software-hardware co-design. It uses PPO with joint search space: model accuracy and hardware constraints.

##########################################################################

Reasons for score:

The paper claims it demonstrate effectiveness of hardware aware NAS for first time, which is dubious.

The paper should reference the industry-standard parametrized accelerator used. And reference or briefly present the in-house simulator.

In the beginning of section 3, it is unclear about the entire workflow. Explain the high-level workflow in Figure 1 caption.

Tested only on Imagenet, mobilnet and efficientnet. It didn't demonstrate that the method can be generalized across different the vertical stack as claimed in the introduction.

In 1. introduction, typo: Intel’s Nervana

##########################################################################

Pros:

- Demosntrated improved accuracy and latency upon other work

##########################################################################

Cons:


- The paper needs to improve its clarity by explaining the figure workflow in section 3.

- It also needs to revise the claim that it demonstrate effectiveness of hardware aware NAS for first time. E.g: NSGA-NETmultiobjective

##########################################################################

Questions during rebuttal period:


Please address and clarify the cons above

---

### Official Review · AnonReviewer1 · 2020-10-28
**Official Blind Review #1**

**Rating:** 4
**Confidence:** 4

**Review:**

This paper proposes NAHAS for co-designing neural network architecture and hardware architecture. It shows performance improvements on ImageNet compared to previous hardware-aware NAS methods which only optimize the neural network architectures.

Pros:
1. Co-designing neural network architecture and hardware architecture is a promising and important direction.
2. The proposed method clearly outperforms hardware-aware NAS.

Cons:
1. The technical contribution of this paper is limited. The main difference between this paper and previous hardware-aware NAS papers is that this paper has an additional hardware search space beside the neural network architecture search space. The other components, such as the training method, search algorithm, and learning objectives, are borrowed from previous papers with minor modifications.

2. Co-designing neural network architecture and hardware architecture is not new. Similar ideas have been explored in [1]. The authors should discuss the difference between this paper and [1].

In summary, co-designing neural network architecture and hardware architecture is not new. The authors do not discuss the difference between this paper and [1]. Besides, I think the technical contribution of the proposed method is limited. Therefore, I recommend rejecting this submission.

[1] Neural-Hardware Architecture Search, NeurIPS 2019 Workshop on Machine Learning for Systems.

---

### Official Review · AnonReviewer4 · 2020-10-28
**NAHAS review**

**Rating:** 5
**Confidence:** 3

**Review:**

The authors describe an automated system for co-designing neural architectures and HW accelerators. The system is able to find the best solution under latency and chip-area constraints.  A highly parameterized (commercial) edge accelerator defines the hardware search space.  Results are compared to MnasNet, platform-aware NAS and EfficientNet.

This is an interesting area and the authors demonstrate clear advantages of their approach.

A claim is made that "although previous works have optimized either neural architectures given fixed hardware, or hardware given fixed neural architectures, none has considered optimizing them jointly.". Does the work outlined in the "co-design" paragraph of section 2 not attempt to do this?  Some other examples that could have been discussed include:

* “FPGA/DNN co-design: An efficient design methodology for IoT intelligence on the edge”, Hao et. al. DAC’19
* “Best of Both Worlds: AutoML Codesign of a CNN and its Hardware Accelerator”, Abdelfattah et al, DAC’20
* When Neural Architecture Search Meets Hardware Implementation: from Hardware Awareness to Co-Design
https://ieeexplore.ieee.org/abstract/document/8839421

It is a little unclear why the papers by Jiang and Yang and the papers above are dismissed? It would be good to understand better how the author's NAS approach improves on these previous works?

Limited information is provided regarding the architecture of the accelerator. What fraction of the HAS search space is unavilable due to restrictions imposed by the compiler? Perhaps these design points are all uninteresting?

Is there anything to learn by a discussion of the good hardware configurations that were found? Are these surprising or unexpected? How do they differ from the baseline configuration?

The work is interesting but the claimed contributions perhaps need to be clarrified.

---

### Official Review · AnonReviewer3 · 2020-10-29
**Good exploration with weak technical contributions.**

**Rating:** 5
**Confidence:** 2

**Review:**

##########################################################################

Summary:

The paper proposes an algorithm to search for hardware designs and neural architectures jointly.
The results show that joint optimization improves accuracy and latency, reduces the search samples.

##########################################################################

Pros:

- The paper introduces a new dimension, hardware design, to the neural architecture search domain.
- The results show that joint optimization outperforms phased optimization.

##########################################################################

Cons:

- The used techniques are not novel
- The hypothesis "joint optimization is better than two-phased optimization" seems obvious.

##########################################################################

Other comments:

1. Can you describe more of the hardware. Is it a publicly accessable platform? How long does it take to switch between different configurations?
2. Can you add wall-clock time or used computing resource in table 3?
3. I also have concerns about the motivation. Is it typical to design an accelerator for just one network? I think an accelerator should be optimized for at least a range of neural networks.

typo: Section 3.1 "The objective for NAHAS it to" -> "The objective for NAHAS is to"

---

### Author Response · Authors · 2020-11-19
**Authors' responses to all reviewers**

We thank the reviewers for their feedback. We would like to address some common questions:

**Q1.** NAHAS novelty compared to related work?

**A1.** To our best knowledge, NAHAS is the first work on co-optimizing neural architectures and hardware accelerators based on industry-standard accelerators and large-scale workloads such as ImageNet. All the hardware performance metrics in our paper were generated using a cycle-accurate simulator, taking a binary code generated by a machine learning compiler. The binary code contains rich information of the program and the target device that enables a more accurate evaluation of the hardware metrics. However, all three related works recommended by Rev#2 are targeting FPGAs with smaller search spaces, where FPGAs are usually used for prototyping ideas. The compilers and target hardware are less optimized compared to an industry-standard accelerator like the NAHAS used. Therefore, our numbers presented in this paper are built on top of much stronger baselines compared to related work. In addition, none of the related work has demonstrated strong empirical results on large, realistic benchmarks like ImageNet, compared to SoTA models, such as EfficientNet. We would consider improvements over a stronger baseline much more significant, compared to related work.

|               Paper              |             Hardware                    |  Large-scale Data | SoTA Accuracy  |
|:-------------------------------:|:---------------------------------------:|:-----------------------:|:----------------------:|
| FPGA/DNN Co-Design | FPGA                                       |              No          |   No |
| Best of Both Worlds      | FPGA                                       |              No          |   No |
| NAS Meets Hardware   | FPGA                                       |              No          |   No |
| **Our NAHAS**             | industry-standard accelerator  |       **Yes**          |  **Yes** |


**Q2.** Should the accelerator be optimized over a range of applications?

**A2.** We agree that multi-task optimization would make the hardware design more generalizable to unseen tasks and evolving workloads. However, we would like to emphasize the importance of applications using jointly optimizing a single-task. It is quite common that in a mobile systems-on-chip environment, a set of accelerators (around 20 accelerators) are designed for different workloads. Moreover, for real-time workloads like foreign object detection on autonomous driving vehicles, it is highly desirable to co-optimize the model and the hardware accelerator to meet the stringent latency objective. NAHAS has demonstrated strong empirical results in reducing latency while improving model accuracy and chip area. We can evision that in the future, the reduction of costs in chip design and manufacturing further necessitates the optimization of accelerators over narrow downed domains. This paper is the first step towards that end goal and we will investigate multi-task co-optimization in our future work

**Q3.** Hardware details are not available.

**A3.** We understand the reviewers’ concern that more hardware details might be helpful.. However, we would like to point out that like many related work [1], the hardware accelerator used in NAHAS is a highly parameterized design that all the important tunable parameters are listed in Table 1. We include most of the important details in Section 3.3 and Section 4.1, that is essential for reproducing the results. Our method should not be limited to one particular hardware accelerator, it can be applied to any parameterized hardware design, like the hardware design in related work [1]. The proposed unified search space joining NAS and HAS, efficient oneshot search with a cost model, analysis over joint search and phase search are practical solutions and insights that should generalize to different hardware platforms. Unlike most related work, our target hardware is optimized over a set of highly competitive industry workloads and the performance simulator is validated, which makes our improvement even more significant compared to results generated on less optimized hardware.

[1] Neural-Hardware Architecture Search, NeurIPS 2019 Workshop on Machine Learning for Systems.

---

### Decision · Program_Chairs · 2021-01-07
**Final Decision**

**Decision:**

Reject

**Comment:**

This paper considers the problem of searching over the joint space of hardware and neural architectures to trade-off accuracy and latency.

Reviewers raised some valid questions about the following aspects:
1. Low technical novelty
2. Prior work on hardware and neural architecture co-design, and closely related work are not addressed
3. Lacking details on hardware platform and discussion on physical constraints to determine invalid hardware designs (addressed somewhat, but the response is not satisfactory)

One additional comment: if we care about latency for a particular hardware platform, it is possible to automatically configure adaptive inference techniques to meet the latency constraints.

Overall, my assessment is that the paper requires more work before it is ready for publication.